# New *N*-Adducts of Thiadiazole and Thiazoline with Levoglucosenone and Evaluation of Their Significant Cytotoxic (Anti-Cancer) Activity

**DOI:** 10.3390/cancers16010216

**Published:** 2024-01-02

**Authors:** Tomasz Poplawski, Grzegorz Galita, Joanna Sarnik, Anna Macieja, Roman Bielski, Donald E. Mencer, Zbigniew J. Witczak

**Affiliations:** 1Department of Pharmaceutical Microbiology and Biochemistry, Medical University, 92-215 Lodz, Poland; tomasz.poplawski@umed.lodz.pl (T.P.); anna.macieja@umed.lodz.pl (A.M.); 2Department of Clinical Chemistry and Biochemistry, Medical University, 92-215 Lodz, Poland; grzegorz.galita@umed.lodz.pl; 3Department of Rheumatology, Medical University, 90-050 Lodz, Poland; joanna.sarnik@umed.lodz.pl; 4Department of Pharmaceutical Sciences, Nesbitt School of Pharmacy, Wilkes University, 84 W. South Street, Wilkes-Barre, PA 18766, USA; bielski1@verizon.net; 5Department of Chemistry and Biochemistry, Wilkes University, 84 W. South Street, Wilkes-Barre, PA 18766, USA; donald.mencer@wilkes.edu

**Keywords:** levoglucosenone, thiadiazole, thiazoline, apoptosis, DNA damage

## Abstract

**Simple Summary:**

The anticancer effects reported in several heterocyclic derivatives of thiadiazole and thiazoline prompted us to construct functionalized new carbohydrate heterocycle entities and investigate their potential anticancer effects on several cancer cells. Here, we show that two novel heterocyclic moieties linked to functionalized carbohydrate with an anhydro scaffold skeleton caused the death of the investigated cancer cells. Interestingly, both *N*-adducts augmented the cytostatic effect on several cancer cell lines with totally different mechanisms of action. The thiazoline derivative responds via apoptosis through caspase activation and the thiadiazole derivative responds through oxidative stress, DNA damage, and necrosis at the micromolecular level.

**Abstract:**

The conjugate N-adducts of thio-1,3,4-diazole and 2-thiazoline with levoglucosenone were synthesized via a stereoselective, base-catalyzed conjugate N-Michael addition to levoglucosenone at C-4. Structural assignments were established using 1H and 13C NMR analysis, and X-ray single-crystal analysis for one of the compounds. The biological properties of the novel compounds were tested on a cell model. Cytotoxicity was analyzed via colorimetric assay. Two distinct types of cell death, apoptosis and necrosis, were analyzed by determining the phosphatidylserine levels from the outer leaflet of the plasma membrane, caspase activation, and lactate dehydrogenase release. We also evaluated DNA damage using an alkaline comet assay. The level of oxidative stress was measured with a modified comet assay and an H2DCFDA probe. The thio-1,3,4-diazole adduct (FCP23) and the 2-thiazoline adduct (FCP26) exhibit similar cytotoxicity values for cancer cells (ovarian (A2780), breast (MCF-7), cervix (HeLa), colon (LoVo), and brain (MO59J and MO59K)), but their mechanism of action is drastically different. While FCP23 induces oxidative stress, DNA damage, and necrosis, FCP26 induces apoptosis through caspase activation.

## 1. Introduction

The so-called war on cancer has had limited success for many years. However, in the past two decades or so, substantial progress has been made on this front. The introduction of earlier detection methods and the addition of various immunotherapies to classical treatments have dramatically improved the outcomes of many cancers. Nevertheless, there is always an acute need for novel, effective molecules that are capable of selectively killing cancer cells. Small molecules belonging to carbohydrate derivatives are such potential candidates. Specifically, we decided to study the anticancer properties of selected Michael addition products to levoglucosenone.

The introduction of the “sugar code” [1] has led us to propose the new concept of the design and synthesis of sugar derivatives with unique chemical properties as Functional Carb-Pharmacophores (FCPs) [2]. We have synthesized various thio sugars using the “thio-click” approach and tested their biological properties. Our collection of FCPs includes monosaccharides containing hemi-thioketal functionality, where the sulfur atom comprises a part of the ring and thio sugar derivatives with a sulfur atom as a bridge connecting various motifs [2,3]. The most interesting FCP, with both strong anticancer and antimicrobial potential, is (1–4)-*S*-thiodisaccharide, which has two sugar moieties linked by a sulfur bridge (named FCP6) [4,5] (Figure 1).

It was shown that FCP6 induces both oxidative stress and endoplasmic reticulum stress related to impaired thiol-dependent cellular pathways responsible for maintaining redox homeostasis in cancer cells [6]. Recently, we expanded our FCP collection with two novel compounds. They result from the Michael addition of the 1,3,4-thiadiazole and 2-thiazoline moieties to levoglucosenone. We named these compounds FCP23 and FCP26, respectively (Figure 1). 

We chose to introduce thiadiazole and thiazoline ring systems, since compounds containing such moieties are known to exhibit biological activities, including anticancer, antimicrobial, antiviral, antidiabetic, and anti-inflammatory properties [4,5,6].

## 2. Materials and Methods

### 2.1. Synthesis of FCP23 and FCP 26

To introduce thiadiazole moiety into the position C-4 of a sugar scaffold, we followed the same protocol as the one we previously developed [5,6] for the synthesis of *S*-linked heterocyclic derivatives via the thia-Michael addition of thiols to levoglucosenone 1.

As shown in Figure 2, both substrates 1,3,4-thiadiazole **2** and 2-thiazoline **3** exist as combination of tautomers. It seems that the thionotautomers exclusively produce both N-adducts FCP23 and FCP 26 (Figure 3). 

The Michael addition of 1-thio-1,3,4-thiadiazole (**2**) to levoglucosenone **1** produced crystalline adduct FCP23 in a good 65% yield. The ^13^C spectrum revealed the presence of two signals, which we attributed to C=S function at 158.95 and C=O at 177.52 ppm, respectively.

Analogously, the conjugate Michael addition of 2-thio-thiazoline **3** to levoglucosenone **1** produced a crystalline *N*-adduct FCP26 in a good 74% yield. The structure of the products was determined using ^1^H, ^13^C NMR and MS.

Additionally, FCP26 with a 2-thiazoline motif linked to the sugar moiety via the nitrogen bridge was studied using single-crystal X-ray crystallography. The ORTEP diagram is presented below (Figure 2). 

We concluded that, in both cases, the *N*-adduct is formed as the only product. There are a few literature reports describing similar results, i.e., Michael addition to **1** giving only *N*-adduct, but no *S*-adduct [7,8].

For example, Spanevello et al. [7] used 2-benzothiazole as reactant in the conjugate Michael addition to levoglucosenone. A European patent [8] reported the synthesis of various *N*-adducts via the conjugate addition of thioxo-heterocycles to levoglucosenone under basic conditions. We observed comparable results in the conjugate addition of thiouronium salt to levoglucosenone [9] when the *N*-imino tautomer undergoes stereoselective conjugate addition at C-4 with concomitant cyclization with C=O at C-2. 

Additionally, Greatrex and coworkers [10] reported the aza-Michael conjugate addition to levoglucosenone under aqueous conditions. 

In order to verify the molecular and biological properties, we performed an in silico analysis by calculating simple molecular descriptors using the Molinspiration online property calculation tool kit (http://www.molinspiration.com, accessed on 15 November 2023).

The Molinspiration Galaxy3D structures of FCP23 and FCP26, (Figure 3) as generated below, clearly show the differences in the molecular properties. Specifically, the distance of the S=O functionality in the thiadiazole ring from the C=O functionality of the 1,6-ahydro ring is longer in FCP23 in comparison to FCP26. We speculate that this is probably the main reason for activity differences during the binding of both functional groups to the proteins in various cancer cells.

The calculated data of the molecular descriptors are listed in Table 1.

According to Lipinski’s [11] rule, five potentially orally active drug candidates should have (I) no more than five hydrogen bond donors, (II) no more than 10 hydrogen acceptors, (III) the octanol/water partition coefficient (logP) should not be >5, and (IV) should have a molecular weight <500 Da. The rotatable bound count (n-Rotb) and topological polar surface area (TPSA) were also calculated for the thiadiazole and thiazoline N-adducts. The numerical values presented in Table 1 are indicative of the better biological activity of thiadiazole and provide additional verification of our experimental results.

Both heterocyclic scaffolds FCP23 and FCP26 represent classic geminal diheteroatomic motifs [12] in which two heteroatoms, oxygen and oxygen (of the acetal), are bound to a single sp3 carbon atom (marked in red in Figure 4). In contrast, the nitrogen and sulfur atoms of thiadiazole and thiazoline combinations comprise a sp2 carbon atom bonded to two S atoms and an N atom, as diheteroatomic motifs are bound to a different carbon atom functionalized with a thioxo motif. This would presumably create the diverse biological activity of FCP23 and FCP26.

### 2.2. Analysis of Biological Properties

#### 2.2.1. Cell Lines

Ovarian (A2780), breast (MCF-7), cervix (HeLa), colon (LoVo), and brain (MO59J and MO59K) cell lines were purchased from the American Type Culture Collection (ATCC; Rockville, MD, USA), except A2780 (Sigma-Aldrich, St. Louis, MO, USA). The growth mediums were supplemented with glutamine (Lonza, Walkersville, MD, USA), 10% fetal bovine serum (BioWhittaker, Lonza, Walkersville, MD, USA), and 1% penicillin–streptomycin mix (Sigma-Aldrich, St. Louis, MO, USA) of the total volume (*v/v*). Human prostate normal PTEN2 cells were cultured in RPMI 1640 (Sigma) medium supplemented with penicillin–streptomycin mix (Sigma), 2 mM glutamine (Sigma), and fetal bovine serum (BII). The cells were gently harvested using Accutase (Sigma-Aldrich, St. Louis, MO, USA). All incubations were processed in a humidified incubator at 37 °C, 5% CO_2_. The cells were passed every 2–3 days at 80% confluence. 

#### 2.2.2. Biological Tests

FCP23 and FCP26 were soluble under experimental conditions and the final concentration of DMSO in the analyzed samples did not exceed 0.5% in the culture medium. All controls described in this manuscript also received 0.5% DMSO. The cytotoxicity of FCP23 and FCP26 was examined as previously described with the colorimetric assay of the Dojindo Cell Counting Kit-8 (CCK-8) (Sigma-Aldrich, St. Louis, MO, USA) [13]. The CCK-8 kit is based on the WST-8 reduction in tetrazolium salt through mitochondrial dehydrogenases that lead to the production of yellow formazan. The intensity of the color is proportional to the cell viability. The cells were cultured in 96-well tissue culture plates for 24 h before the experiment, allowing them to adhere to the surface. The cells were washed with PBS and supplemented with fresh medium containing a 10% CCK8 solution after 72 h of incubation with FCP in the concentration range from 0 µM to 200 µM. After 3 h of incubation, the absorbance was measured at 450 nm using a Synergy HT microplate reader (BioTek Instruments, Winooski, VE, USA). The IC_50_ was calculated using The Quest Graph™ IC50 Calculator (“Quest Graph™ IC50 Calculator.” AAT Bioquest, Inc., Pleasanton, CA, USA, https://www.aatbio.com/tools/ic50-calculator, 21 December 2023). It models an experimental set using a four-parameter logistic regression model.

We also analyzed the main signal of apoptosis: the activation of executioner caspases 3/7 with the homogeneous caspase-3/7 assay (Promega, St. Louis, MO, USA). Necrosis was analyzed by measuring the release of lactate dehydrogenase (LDH) and the translocation of phosphatidylserine from the inner leaflet to the outer leaflet of the plasma membrane using the FITC Annexin V assay (Sigma-Aldrich, St. Louis, MO, USA).

DNA damage was tested with the alkaline version of the comet assay. The comet assay is a rapid and sensitive assay for analyzing the level of DNA lesions such as single- and double-stranded damage and alkali labile sites. The use of human 8-hydroxyguanine DNA-glycosylase (hOGG1) in the modified comet assay allowed us to detect 8-hydroxy-2-deoxyguanine, a known marker of oxidative DNA damage [13,14]. We also estimated the level of reactive oxygen species (ROS) induced by the compounds tested using an H2DCFDA probe. The test is based on the oxidation of the non-fluorescent H2DCFDA compound to the highly fluorescent 2’,7’-dichlorofluorescein (DCF) by ROS. The amount of DCF is proportional to the amount of ROS generated by the compound.

## 3. Results and Discussion

The rationale for the synthesis of the selected compounds is the potential influence of thiadiazole and thiazoline, heterocyclic moieties, on the anticipated anticancer properties. Thiadiazoles, a subfamily of azoles, are used as bioisosteres of thiazole, oxazole, oxadiazole, and benzene for the development of analogs with biological properties comparable to those of their precursors. Their reactivity arises from the presence of the nucleophilic center located in the ring, and the electrophilic center on the carbon atom of the C=N bond. Their use as functional moieties in drug design demonstrates a wide spectrum of actions, including antimicrobial, antifungal, antioxidant, and antidepressant activities [15,16,17]. 

Thiazoline, a reduced form of thiazole, is another significant motif that belongs to a class of potential antitumor drugs and has an inhibitory effect on DNA synthesis [18]. Our experimental setup is based on our previously reported scheme [2]. This includes a screening cytotoxicity analysis aimed at selecting the most responsive cancer cell line and a selection of compounds with anticancer potential for further investigation.

First, we evaluated the cytotoxic activities of FCP23 and FCP26 against six models of human cancer cells: ovarian (A2780), breast (MCF-7), cervix (HeLa), colon (LoVo), and brain (MO59J and MO59K). We chose the certain concentration range of FCPs because it fits the NCI recommendations for a potential anticancer drug and is also similar to the concentration array used in our previously published studies, and we expected similar bioactivity [2].

Based on the results presented in Table 2, we decided to perform all subsequent experiments with the A2780 cell line.

FCP26 exhibited the highest cytotoxic activity (IC_50_ 47.8 µM). Replacing the thiazoline moiety with the aminothiadiazole moiety (FCP23) did not increase the cytotoxicity of the compound (IC_50_ 50.2 µM). Note that both compounds were cytotoxic within micromolar IC_50_ values. The IC_50_ values were similar to those obtained for the known anticancer drug etoposide. Representative figures demonstrating the activity dependence of the varied concentrations of the compounds, along with the approximation curves, are presented in the Appendix A section. We also found that normal cells are more resistant to the studied compounds compared to cancer cells (Appendix A). We used human prostate normal PNT2 cells, immortalized with SV40, as they showed similar characteristics to A2780 ovarian cancer cells (adherent, epithelial).

Next, we evaluated the apoptosis and necrosis potential of the compounds studied. We observed a striking difference between FCP23 and FCP26 (Figure 4A–C). We observed signals for the presence of phosphatidylserine in the outer leaflet of the plasma membrane when the cancer cells were incubated with both compounds (Figure 4A). More phosphatidylserine was present for FCP23 (32% at IC_20_ and 86% at IC_50_), and the amount of phosphatidylserine for FCP26 was significantly lower (10% at IC_20_ and 22% at IC_50_). In contrast to FCP23, FCP26 induced apoptosis, since we observed higher caspase 3/7 activity in cells incubated with FCP26 than those incubated with FCP23 (Figure 4B, 4.3 ± 1.2 at IC_20_ and 15 ± 2.1 at IC_50_ vs. 0.5 ± 0.1 and 1 ± 0.3). The situation changed when we analyzed LDH activity. In this particular scenario, LDH activity was observed to have a larger effect for FCP23 (Figure 4C, 23.8 ± 2.9 at IC_20_ and 62 ± 5.4 at IC_50_ vs. 5.6 ± 1.7 and 15.9 ± 4.8). These data suggest that FCP23 causes necrosis, while FCP26 instead induces apoptosis through caspase activation.

In the next step, we analyze the potential of the tested compounds to induce DNA lesions using the alkaline and modified version of the comet assay (Figure 5). 

The comet assay is a rapid and sensitive assay for analyzing the level of DNA lesions such as single- and double-stranded damage and alkali labile sites. The use of human 8-hydroxyguanine DNA-glycosylase (hOGG1) in the modified comet assay allowed us to detect 8-hydroxy-2-deoxyguanine, a known marker of oxidative DNA damage [14]. Cancer cells were incubated with FCP23 at 14 µM or FCP26 at 12 µM within 120 min. The level of DNA damage was measured at five time points over a period of 0 min to 120 min. Both compounds caused DNA lesions, as shown in Figure 5A; however, the level of damage evoked by FCP23 was three times higher than that evoked by FCP26 (median 21% vs. 7%). Furthermore, the genotoxic effect of FCP23 accumulated over time and the amount of DNA damage exceeded the 20% threshold in 120 min, while FCP26 caused a DNA damage ratio of 10% all the time. We also observed a several-fold increase in the level of DNA damage ratio after hOGG1 treatment (Figure 5B). This effect was only characteristic for the comet test samples obtained after the incubation of A2780 cells with FCP23, but not FCP26. This observation suggests that FCP23 induced oxidative DNA lesions in contrast to FCP26. To verify this hypothesis, A2780 cells were incubated with FCP23 or FCP26 in the presence and absence of two different free radical scavengers: tocopherol and PBN. In fact, both tocopherol and PBN significantly reduced the DNA damage induced by FCP23. This effect was not seen for FCP26 (Figure 5C). Next, we estimated the level of reactive oxygen species (ROS) induced by the compounds tested using the H2DCFDA probe. The test is based on the oxidation of the non-fluorescent H2DCFDA compound to the highly fluorescent 2′,7′-dichlorofluorescein (DCF) by ROS. The amount of DCF is proportional to the amount of ROS generated by the compound. The results show that FCP23, in contrast to FCP26, generated a significant increase in ROS (*p* > 0.001, Figure 6).

Although the structures of the newly synthesized compounds are relatively similar, their biological properties are completely different. Interestingly, both compounds exhibit similar cytotoxicity values, but their mechanism of action seems to be drastically different. FCP23 induces oxidative stress, while FCP26 does not. These results suggest that FCP23 acts in a manner similar to FCP6.

There is no proof that the anhydrosugar moiety has any effect on biological properties. Based on our prior research, this motif appears to aid the transfer of FCP into cancer cells, where it is degraded/hydrolyzed under cancer-cell-specific conditions (pH) and enable the thiazoline pharmacophore to interact with intracellular thiols via its sulfur moiety. 

## 4. Conclusions

In conclusion, the mechanisms of biological action of the newly synthesized compounds are strikingly different. Both compounds exhibit similar cytotoxicity for cancer cells, but their mechanism of action seems to be drastically different. FCP23 induces oxidative stress, DNA damage, and necrosis, while FCP26 kills cancer cells via apoptosis. From the point of view of the suitability of these compounds as potential anticancer drugs, FCP26 seems to be better than FCP23. First, it kills cancer cells by inducing the apoptosis process, which is preferred as a mechanism of action for anticancer drugs compared to necrosis. Apoptosis is defined as an activated cellular degradation process that prevents the induction of underlying inflammation, while necrosis has been characterized as a passive cell death resulting from abnormal environmental conditions with the uncontrolled release of inflammatory cell contents. A review of the few available literature data on the anticancer properties of thiazolines suggests that they are potent inhibitors of PIM kinases, a type of serine/threonine kinase abnormally expressed in many cancers [19,20]. Other biological activities of thiadizole and functionalized thiazoline are reported in the literature as well [21,22,23,24,25,26,27,28,29,30,31,32,33,34].

To better understand the biological activity, a structure–activity relationship analysis should be performed that would include the synthesis and biological evaluation of a series of FCP26 analogs bearing different thiazoline moiety. Such research is currently being conducted in our laboratory. The preliminary results were presented in [35] and will be reported elsewhere.

## Data Availability

The data presented in this manuscript are available on request from the corresponding author.

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
