# Peer review of "New N-Adducts of Thiadiazole and Thiazoline with Levoglucosenone and Evaluation of Their Significant Cytotoxic (Anti-Cancer) Activity"

_cancers, 2024, doi:10.3390/cancers16010216_

Round 1

Reviewer 1 Report

Comments and Suggestions for Authors

REVIEWER'S REPORT

Manucsript title: New N-adducts of thiadiazole and thiazoline with levoglu-cosenone and evaluation of their significant cytotoxic (anticancer) activity (Authors: Tomasz Poplawski, Grzegorz Galita, Joanna Sarnik, Anna Macieja1, Roman Bielski, Donald E. Mencer, Zbigniew J. Witczak)

   The authors describe the continuation research of their previous studies in this publication. The conjugated N-adducts of thio-1,3,4-diazole and 2-thiazoline with levoglucosenone were created in their current work, and their anticancer activity against a panel of cancer cell lines was examined. Both compounds had virtually similar cytotoxic effectiveness against the cancer cell lines tested, but further extensive research revealed that their modes of action appear to be distinct. This manuscript, in my opinion, correctly done and written in clear English, and it might be published in this journal. Nonetheless, before it can be accepted, the manuscript should be proofread in certain tex places and supplemented to improve manuscript quality.

I would  advise changing or modifying several passages in the article's text, in particular:

In Introduction. The sentences in lines 39-40 have to be corrected/paraphrased. The significance and objectives of this work must be described in greater detail and clarity at the end of this section.

In Materials and Methods. Subsection 2.1. The sentence in lines 73-47 should be written as "As shown in Scheme 2, both 1,3,4-thiadiazole 2 and 2-thiazoline  molecules exist as a combination of tautomers". As for 1.3,4-thiadiazole, this molecule is likely to have more than two tautomeric forms. Which of them is most stable?  

In Results and Discussion. The sentence in lines 194-195, can be paraphrased as follows " Their use as functional moieties in drug design demonstrates a wide spectrum of actions, including antibacterial, antifungal, antioxidant, antidepressant, and anticancer properties."  In lines 196-, the sentence should be rewritten as " Thiazoline, reduced form of thiazole, is another clinically significant motif that belongs to..." The sentence in lines 253-254 should be paraphrased as " Both compounds caused DNA lesions, as shown in Fig.5A"

The sentence in lines 283-284, should be written as " There is no proof that the anhydrosugar moiety has any effect on biological properties. Based on our prior research, this motif appears to aid in the transfer of FCP..."

In Conclusion. In line 302, instead of "To understand...", should be written "To better understand..."  

 Supplements and other information should be included, specifically: 

1) As for 1.3,4-thiadiazole (Scheme 2), this molecule is likely to have more than two tautomeric forms. Which of them is most stable (experimental or computational justification) ?

  2) The methodology utilised in this work to estimate the CL50 parameter values of the cytotoxic effects of the examined compounds on cancer cell lines is not provided in Materials and Methods. It is unclear how CL50 was calculated, whether empirically or by fitting experimental data with logistic (or other type) equations. Typically, EC50 values are defined through data approximation with logistic equations, yielding CL50 (at inflection points of activity against varied concentrations of the compounds). To my mind, in this work, one or more Figures demonstrating the activity dependence of the varied concentrations of the compound(s) along with the approximation curves should be represented.

    3) A control  should be run to assess cytotoxic activity of the compounds examined in this work upon one or more healthy cell lines.  In addition, one or more standard antitumor drugs should be applied as a control.

Comments on the Quality of English Language

The article needs to be corrected in a few places with proper English.

Author Response

Response to referee # 1.

The following changes marked in yellow were introduced into manuscript:

Line 38-40;

We added the following sentence

“small molecules belonging to carbohydrate derivatives, are such potential candidates. Specifically, we decided to study the anticancer properties of selected Michael addition products to levoglucosenone”

Line 72-73; changed to

“ As shown in Scheme 2, both 1,3,4-thiadiazole  2 and 2-thiazoline  3 molecules exist as a combination of tautomers. It seems that the thiono tautomers produce exclusively both N-adducts FCP23 and FCP 26 (scheme 3).

Additional answer: Since cancers is not a chemical journal we decided to draw only essential tautomeric forms.

Line 194-195; changed to

“ Their use as functional moieties in drug design demonstrates a wide spectrum of actions, including antibacterial, antifungal, antioxidant, antidepressant, and anticancer properties”

Line 196-7; changed to

“ Thiazoline reduced form of thiazole, is another clinically significant motif that belongs to a class of potential antitumor drugs.

Line 253-254; changed to

“ Both compounds caused DNA lesions, as shown in Fig.5A”

Line 283-284; changed to

“ there is no proof that the anhydrosugar moiety has any effect on biological properties. Based on our prior research, this motif appears to aid in the transfer of FCP into cancer cells”

Line 302; changed to

“ To better understand..”

Comment: The methodology utilised in this work to estimate the CL50 parameter values of the cytotoxic effects of the examined compounds on cancer cell lines is not provided in Materials and Methods. It is unclear how CL50 was calculated, whether empirically or by fitting experimental data with logistic (or other type) equations. Typically, EC50 values are defined through data approximation with logistic equations, yielding CL50 (at inflection points of activity against varied concentrations of the compounds). To my mind, in this work, one or more Figures demonstrating the activity dependence of the varied concentrations of the compound(s) along with the approximation curves should be represented.

Answer: We have added the following fragment in the end of first paragraph of Biological tests section: IC50 was calculated using The Quest Graph™ IC50 Calculator ("Quest Graph™ IC50 Calculator." AAT Bioquest, Inc., 21 Dec. 2023, https://www.aatbio.com/tools/ic50-calculator). It models an experimental set using a four parameter logistic regression model. 

and in Results section:

Representative figures demonstrating the activity dependence of the varied concentrations of the compounds along with the approximation curves are represented in Supplementary Section.

Comment: A control  should be run to assess cytotoxic activity of the compounds examined in this work upon one or more healthy cell lines.  In addition, one or more standard antitumor drugs should be applied as a control.

Answer: In our studies (PMID: 25466184 DOI: 10.1016/j.bmcl.2014.10.095) we usually use human prostate normal PNT2 cells, immortalized with SV40, as they showed similar characteristics to cancer cells (including A2780 cells – adherent and epithelial).We apologize that we did not include this results in first version of manuscript. We found that normal cells are more resistant to studied compounds as compared to cancer cells. Standard antitumor drugs used for treatment of ovarian cancer are cisplatin and etoposide. We have research experience with both drugs (doi: 10.3390/cimb45100500. PMID: 37886943; PMCID: PMC10605129) and according to reviewer suggestion  we decided to include the results of etoposide's cytotoxicity to A2780 cells in this paper. Etoposide shows a slightly stronger cytotoxic effect than the studied compounds. We have added this information to the results and graphs in supplementary sections.

Reviewer 2 Report

Comments and Suggestions for Authors

cancers-2784749

This manuscript describes the preparation of two new levoglucosenone derivatives, which were tested against a panel of cancer cell lines and then, their properties as apoptotic, necrotic and oxidative stress inducing factors was examined.

It is a well written manuscript, the chemistry is straightforward, the identification of the new compounds is complete and the selectivity of the Michael addition is interesting. I think that the discussion on Lipinski’s rules is overextended and does not offer much to this paper. I would suggest to reduce this part (Lines 118-135). The cytotoxicity of the two compounds against the six cell lines tested is not impressive (upper micromolar levels) and a corresponding statement could be inserted into the text. On the contrary, the interesting biological finding is the difference on the behavior of the two compounds on necrosis or apoptosis potential, as well as on ROS induction and this is worth to be emphasized.

Some point to be taken under consideration are listed below.

Line 73. Please insert compound numbers in parentheses when complete chemical name is provided: “1,3,4-thiadiazole (2), and 2-thiazoline (3),”. Follow this in any other case into the text (ex. Line 77, etc).

Line 196: “reduced form of thiazole”. In the same paragraph, thiazoline itself, does not belong to antitumor drugs, maybe a different expression should be used.

Lines 278, 289: “of the newly synthesized”

Line 300: the statement “they are potent inhibitors of PIM kinases” is highly speculative, since no relative experiments have been made in this work. I would suggest to point this in the text.

Comments on the Quality of English Language

The quality of English language is good.

Author Response

Response to referee # 2.

All new changes are marked in yellow

Line 73; changed to

“1,3,4-thiadiazole (2) and 2-thiazoline (3)’ 

Line 77 changed to

“ 1,3,4-thiadiazole (2)

Line 118-135 changed to

 “reduced form of thiazole"

Line 196 changed to

“reduced form of thiazole"

Line 278-279  changed to

“ of the newly synthesized”

Line 300 changed to“few available literature data” stating the literature data of thiazolines on potent inhibitors of PIM kinases 

Line 306-307 

the following sentence was added to accommodate corrected new number of references 

Other biological activity of thiadizole and functionalized thiazoline are reported in the literature as well [21-30].